# Beware of the Iceberg Phenomenon: A Case Report of Chest Wall Fibrous Dysplasia

**DOI:** 10.3390/diagnostics14171873

**Published:** 2024-08-27

**Authors:** Vincent van Suylen, Rienhart F. E. Wolf, Wobbe Bouma, Caroline Van De Wauwer, Albert J. H. Suurmeijer, Massimo A. Mariani, Theo J. Klinkenberg

**Affiliations:** 1Department of Cardiothoracic Surgery, University Medical Center Groningen, University of Groningen, 9700 RB Groningen, The Netherlands; rfewolf@gmail.com (R.F.E.W.); w.bouma@umcg.nl (W.B.); c.van.de.wauwer@umcg.nl (C.V.D.W.); m.mariani@umcg.nl (M.A.M.); t.j.klinkenberg@umcg.nl (T.J.K.); 2Department of Pathology, University Medical Center Groningen, University of Groningen, 9700 RB Groningen, The Netherlands; a.j.h.suurmeijer@umcg.nl

**Keywords:** fibrous dysplasia, chest wall tumor, imaging follow-up, hemi-clamshell surgery

## Abstract

Thoracic fibrous dysplasia (FD) is a benign, osseous chest wall tumor. It originates from bone marrow and accounts for 30–50% of all benign osseous neoplasms in the chest wall. In FD, normal bone marrow is replaced by fibrous stroma and immature bone. We present a rare case in which massive intrathoracic polyostotic FD originating from the rib was diagnosed and treated. The extrathoracic part of the tumor appeared stable and unalarming for decades; however, in hindsight, the intrathoracic part significantly progressed, eventually leading to symptoms. The tumor was removed through a hemi-clamshell approach, which allowed adequate visualization and control of mediastinal structures. After establishing the diagnosis of FD, regular follow-up imaging is crucial for timing of a surgical intervention to prevent symptoms, impairment of quality of life, and unnecessarily complex resections.

## 1. Introduction

Primary chest wall tumors account for 2% of all primary tumors, with the majority being malignant [1]. Fibrous dysplasia (FD) is considered a benign neoplasm, originating from the bone marrow and accounts for 30–50% of all benign osseous neoplasms in the chest wall [2]. Normal bone marrow is replaced by fibrous stroma and immature bone. Monostotic FD is the most common expression, followed by polyostotic FD in 25% of cases [3]. The latter presents most frequently in the femur and tibia, with lower frequency in the ribs [4]. FD can present as a single modality, but it is also associated with several syndromes (e.g., McCune–Albright syndrome (MAS)). In general, patients present in the second or third decade of life, without gender predisposition [3]. Patients are, generally, clinically asymptomatic, although local pain can be caused by pathological fractures or periosteal stretching [3]. Given its rarity and non-specific symptoms, FD is frequently misdiagnosed or diagnosed at a late stage [5]. Extrapolating information from the consensus document of the FD/MAS international consortium, a computed tomography scan is recommended for diagnosing FD [5]. In addition, magnetic resonance imaging (MRI) might help differentiate cysts from FD and might have an advantage over CT when the FD is in close proximity to vascular structures and the heart [6].

A cornerstone of the treatment of FD remains surgery, with bisphosphonates playing a supportive role in management of bone quality. A recent review elaborately describes several additional therapies, of which many have not (yet) found their way into clinical practice [7].

The prognosis of FD, being a benign disease, is good. However, either malignant transformation (1%) or pathological growth compromising other vital structures can impair prognosis [4].

This report describes a highly unusual case of FD that appeared as stable palpable extrathoracic mass over decades. In hindsight, the intrathoracic part significantly progressed over these decades, which ultimately led to symptomatic disease.

## 2. Case Report

A 70-year-old woman with a “non-growing” palpable mass of 10 × 10 cm^2^ on the right side of her chest, presented at the general practitioner. The “non-growing” palpable tumor was reported to have been stable for over more than 20 years, and was therefore considered unalarming. Due to malignant disease among her relatives, but in the absence of pain or dyspnea, she became worried and was referred for a chest X-ray by her general practitioner. The X-ray revealed a large intrathoracic mass. She was referred to a pulmonologist for further diagnosis and treatment. Needle biopsies confirmed the typical histology of FD. From the same biopsies, subsequent Guanine Nucleotide binding protein, Alpha Stimulating activity polypeptide (GNAS) mutation analysis—a valuable marker—in addition to the clinical and histological aspects of FD [8] confirmed the diagnosis. Given the expected complexity of the surgery, asymptomatic disease and age of the patient, bi-annual follow-up was considered most suitable.

She developed complaints of progressive dyspnea and impairment of quality of life after approximately 2 years of follow-up. A chest X-ray was performed and demonstrated progressive growth of the mass together with a displacement of the mediastinum to the left (Figure 1).

A computed tomography (CT) scan revealed three locations of FD: two minor lesions posterolateral in the second and third rib and one ventral in the third rib (Figure 2A), the latter being the most prominent lesion and responsible for the palpable extrathoracic mass (Figure 2B and Figure 3A). Its progression resulted in a mediastinal shift to the left (Figure 2C and Figure 3B) with compression of the superior caval vein and right atrium. Vascular displacement with compression of the right pulmonary artery was noticed, with dorsal displacement of the bronchus (Figure 2D).

Lung function analysis showed no significant abnormalities, which was an additional argument for right-sided cardiac compression as the cause of progressive dyspnea.

Given the recent progression of symptoms due to the mass effect of the FD on the right lung, right atrium and mediastinum, complete surgical resection was considered the only potential curative strategy.

A right hemi-clamshell incision was performed through the fourth intercostal space (Figure 4). The intrathoracic mass was dissected free from the mediastinum, thoracic wall and the surrounding lung tissue. Subsequently, the extrathoracic part was dissected free from the overlying muscles and subcutaneous tissue. Finally, the third rib was cut anterior and posterior of the FD, allowing complete resection (Figure 5A,B).

The dimensions of the resected tumor were 17 × 20 × 18 cm^3^ with a weight of 1949 g. Histology again showed the typical aspect of FD (Figure 6). The right middle lobe contained severe parenchymal hematoma and a deformed lobar bronchus. A lobectomy of the right middle lobe was thus performed. Both the right lower and upper lobe were inflated adequately. The pectoral muscle together with the overlying breast tissue were used for closure of the thoracic wall defect.

Postoperatively, the patient went to the ICU and was extubated after 1.5 h. The postoperative chest X-ray showed a normal hilum and a well expanded right lung. The patient was discharged from the ICU the next day and was discharged from the hospital on the fifth postoperative day. The patient recovered swiftly, with additional standard physical therapy from physiotherapy consultants in the first four days. This was well tolerated. The short-term postoperative period was uneventful with full recovery at first outpatient follow-up three months after surgery. After additional follow-up of more than 4 years, the patient was still asymptomatic and there were no radiological signs of recurrence or progression of the other FD sites.

## 3. Discussion

Our case demonstrates the so-called “iceberg phenomenon” in which the tip is only a fraction of a much larger, underlying problem. Significant diagnostic delay was encountered in this case, with the absence of diagnostic imaging and medical follow-up for many years, which eventually resulted in a more complex surgical procedure.

Chest wall tumors comprise a broad range of diseases—both malignant and benign. FD is rare, benign and patients most often do not present with clinical symptoms. If symptoms do occur, they mainly manifest as pain in the second or third decade of life [3]. Some authors describe that FD affects lung function analysis [9,10]. However, not a significant impairment of lung function, but rather, the mass effect on the heart and mediastinum was thought to be the cause of symptom progression in our case. We hypothesize that the direction of growth of non-infiltrating expanding tumors is dependent on the path of lowest resistance [11]. In this case, the FD encountered the overlying muscles and skin extrathoracically and the pleura and lung intrathoracically. The extrathoracic extension was more limited due to the high resistance of the overlying muscles and skin resulting in a more stable palpable mass. The intrathoracic extension was much more elaborate due to the low resistance of the pleura and lung. This hypothesis may explain the “iceberg phenomenon”.

The standard surgical approach to FD originating from the ribs is a (posterolateral) thoracotomy [9,10,12]. In our case, a thoracotomy would not have been a suitable approach, given the dimensions and stiffness of the FD and its relation to the great vessels and phrenic nerve. A thoracotomy would not have allowed a clear visualization of these vital structures and manipulation would be difficult and perhaps dangerous. Therefore, resection of the FD was performed by a right-sided hemi-clamshell approach. The hemi-clamshell was performed through an L-shaped sternotomy, starting at the sternal notch extending to the level of the intercostal space of the anterior thoracotomy, where it merges with this anterior thoracotomy. Recurrence rate after surgical treatment of FD is rarely reported, with predominantly case reports with short follow-up presented in the literature. For craniofacial FD, a systematic review showed recurrence rates of 4% up to 84%, with radical resections in non-syndromic cases and conservative surgery in syndromic cases, respectively [13]. In our case, the anatomical location facilitated radical resection and recurrence is therefore unlikely to occur. Thus, it might be hypothesized that recurrence is highly dependent on the anatomical location, surgical technique and whether FD occurs in a (non-)syndromic form.

Although FD of the rib is a benign disease, its massive intrathoracic growth can result in displacement of organs and subsequent symptoms, even after decades of asymptomatic follow-up as in this case. The case highlights the importance of adequate follow-up with timely and frequent imaging in order to tackle the “iceberg phenomenon”. A CT scan will often be the initial imaging modality, although recently, the “milk cloud appearance” on magnetic resonance imaging (MRI) was suggested to be an adequate alternative for diagnosing FD [6]. In that study, MRI correlated well with histologically proven FD, especially in long bones. Furthermore, the exact frequency of imaging and the choice of the imaging modality is highly dependent on the location and size of FD and its relation to vital structures. MRI may have an advantage over CT in terms of a better definition of the interface of the mass and its neighboring structures [14], especially when the FD is in close proximity to vascular structures and the heart.

Follow-up in this case is limited to 4 years postoperatively, which prohibits drawing conclusions on even longer follow-up, in particular on the evolution of the non-operated sites of the polyostotic FD.

In conclusion, in thoracic fibrous dysplasia, imaging at regular intervals is crucial for the optimal timing of surgical intervention to prevent symptoms and impairment of quality of life and to prevent unnecessarily complex resections.

## Figures and Tables

**Figure 1 diagnostics-14-01873-f001:**
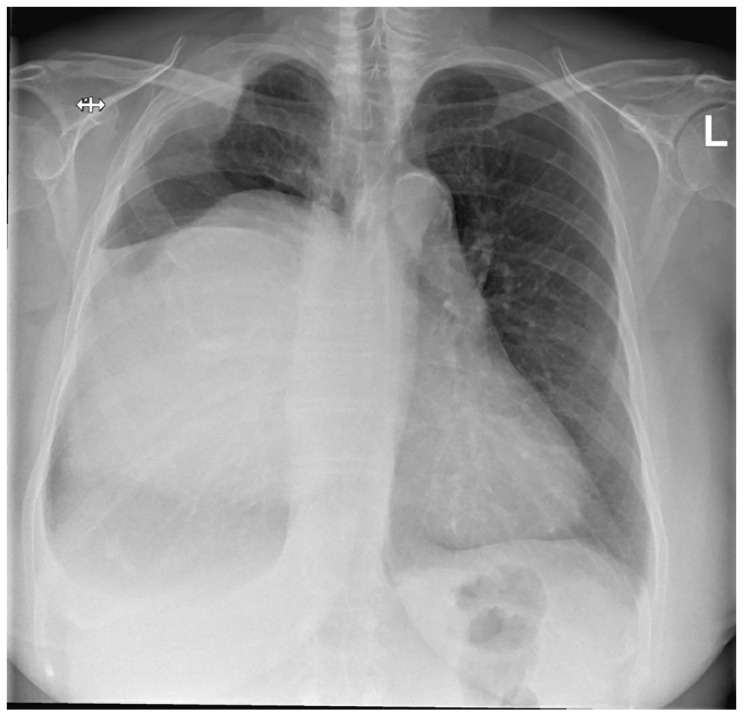
Preoperative posterior–anterior chest X-ray. Evidently increased density of the right hemithorax. The right hemidiaphragm cannot be clearly distinguished, most likely as a result of the large soft tissue mass displacing the mediastinum to the left. In addition, morphological changes of the right lateral side of the second and third rib are visible, with intrathoracic enlargement of the soft tissue.

**Figure 2 diagnostics-14-01873-f002:**
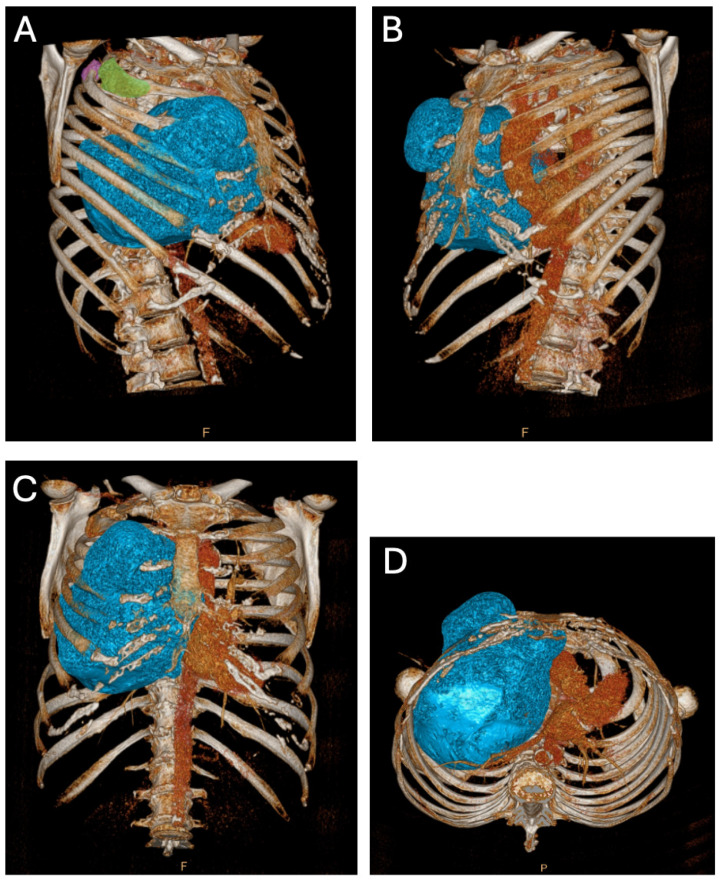
Preoperative three-dimensional reconstruction of a computed tomography dataset. (**A**) Right ventral view. The significance of the major FD originating from the ventral third rib is shown with a blue overlay. The posterolateral FD in the third and second rib are depicted with a purple and green overlay, respectively. (**B**) Left ventral view. The prominent FD resulting from the ventral third rib is depicted with a blue overlay, with focus on the extrathoracic palpable part of the FD. (**C**) Ventral view. Significant mediastinal shift to left. (**D**) Caudal view. The mediastinal shift is clear with posterior vascular and bronchial displacement.

**Figure 3 diagnostics-14-01873-f003:**
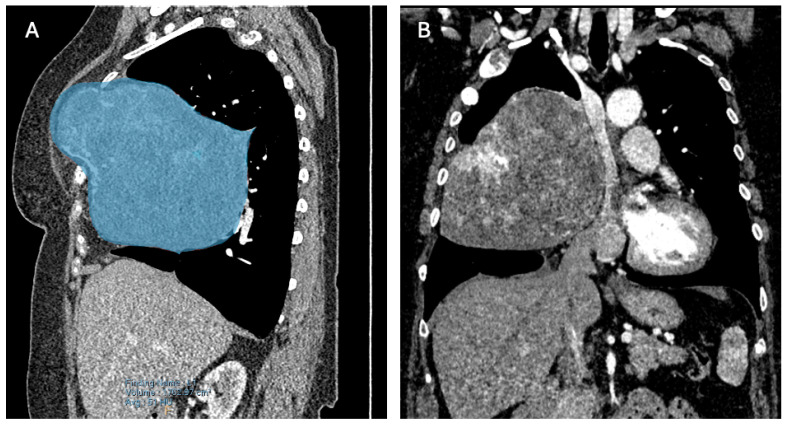
(**A**) A right paramedian multiplanar reformation of the preoperative computed tomography (CT) scan. Blue overlay depicts the extrathoracic palpable FD and its continuity with the greater intrathoracic part. (**B**) A coronal multiplanar reformation of the preoperative CT scan. Visualization of the evident mass effect of the FD.

**Figure 4 diagnostics-14-01873-f004:**
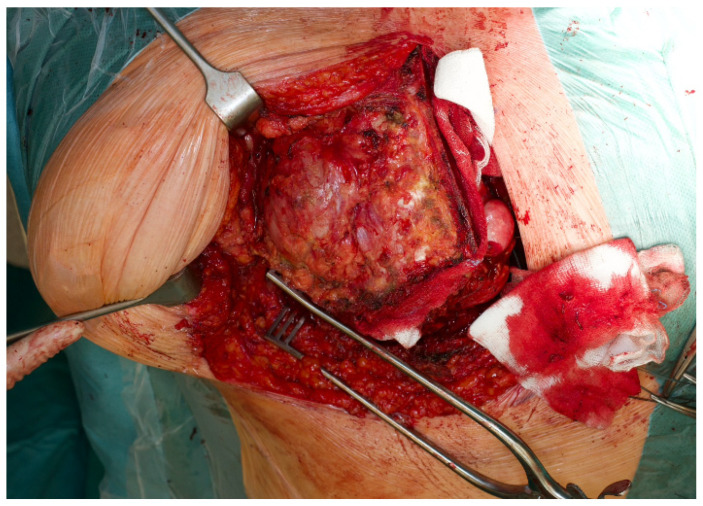
Intraoperative picture of the hemi-clamshell incision, with a clear view on the palpable extrathoracic FD. The partial sternotomy is visible on the right side. The continuity with the thoracotomy is visible on the caudal side.

**Figure 5 diagnostics-14-01873-f005:**
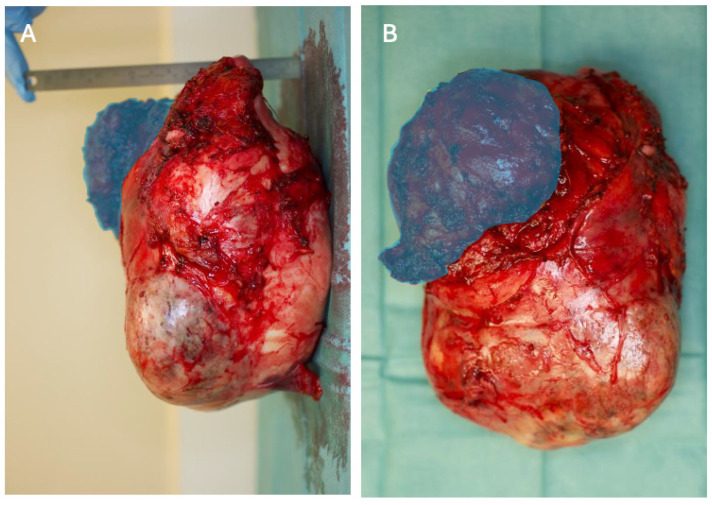
(**A**) Lateral view of the explanted tumor. Figure rotated 90 degrees anticlockwise to be consistent with Figure 3A, the right paramedian multiplanar reformation of the preoperative CT scan. Blue overlay depicts the extrathoracically palpable FD. The dimensions of the tumor were 17 × 20 × 18 cm^3^ with a weight of 1949 g. Right side: dorsal, left side: ventral, top: cranial, bottom: caudal. (**B**) Frontal view of the explanted tumor. Figure rotated 90 degrees anticlockwise to be consistent with Figure 2C, pre-operative three-dimensional reconstruction of a CT dataset, ventral view. Blue overlay depicts the extrathoracically palpable FD. Right side: medial, left side: lateral, top: cranial, bottom: caudal.

**Figure 6 diagnostics-14-01873-f006:**
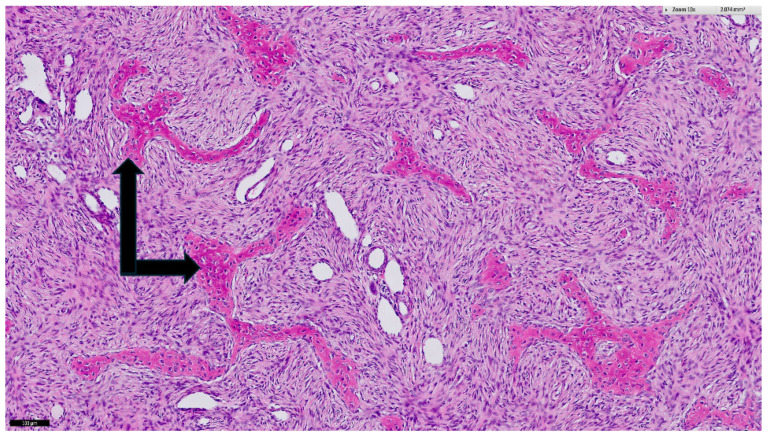
Histological specimen of the resected mass. Hematoxylin and eosin microscopy show the typical morphology of fibrous dysplasia of bone represented by spindle cell stroma harboring irregular trabeculae of woven bone (black arrows) with irradiating collagen fibers. The black bar in the lower left corner corresponds to 100 µm.

## Data Availability

The raw data supporting the conclusions of this article will be made available by the authors on request.

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
