# Peer review of "Beware of the Iceberg Phenomenon: A Case Report of Chest Wall Fibrous Dysplasia"

_diagnostics, 2024, doi:10.3390/diagnostics14171873_

Round 1

Reviewer 1 Report

Comments and Suggestions for Authors

This is a very interesting manuscript. It is well-written and the manuscript describes well the whole pathway from diagnosis to treatment.

I have the following minor suggestions:

1) The introduction should be more extended, providing more information on diagnostic modalities and treatment strategies.

2) A limitations paragraph should be added in the discussion section. 

Author Response

At first, we would like to thank you for your appreciated feedback and useful suggestions. Below you will find the point-to-point responses to your comments. 

Comment 1) The introduction should be more extended, providing more information on diagnostic modalities and treatment strategies.

Response 1) Thank you for pointing this out. We agree that we could give a more elaborate introduction. Therefore, we added sentence 30 up to sentence 47 in the manuscript. We mention the 2019 consensus document on the McCune-Albright Syndrome(MAS)/Fibrous Dysplasia (FD), where they suggest to start with a CT scan when diagnosing MAS/FD. In addition, an MRI could be made to differentiate from cysts.

Additionally, added sentence 41-44, describing the treatment options. The cornerstone remains surgery, with a supportive role for biphosphonates. We refer to a recent review describing potential therapeutical options.

Comment 2) A limitations paragraph should be added in the discussion section

Response 2) We agree. We, accordingly, added the limitation section in the discussion by adding sentence 198 up to 200. We mention the limited follow-up.

Reviewer 2 Report

Comments and Suggestions for Authors

This is a very interesting case that warrants reporting. The surgeons should be commended for their complete resection of the mass using a hemi-clamshell approach. Below are my comments regarding the write-up:

Introduction: The statement about monostotic fibrous dysplasia being the most common form should be followed by specifying the site where it is most frequently reported. Additionally, it would be informative to mention the stage of life at which it most commonly presents.

Case report:

The case report is well-written. However, regarding the figures, it would be beneficial to combine them into sets with figure panels. For example, all 3D constructed CT images could be consolidated into one figure with multiple panels. Similarly, the multiplanar CT images could be grouped. Figures 8 and 9, which show the resected tumor mass, can also be combined.

Please include arrows to point to the spindle cell stroma and irregular trabecular bone in the histopathology slide, preferably in different colors.

Including intra-operative pictures of the hemi-clamshell approach will be interesting for the readers.

The case report ends abruptly without any follow-up data on the patient. How long was the patient followed up? Providing follow-up data for at least 6 months would be beneficial.

It would be useful to include details about the post-operative rehabilitation protocols, including exercises, and how the patient tolerated them.

How long did it take for the patient to return to full function?

Discussion: 

The discussion is concise and appropriately focuses on the findings unique to this case.

If only a minimum follow up is available, it should be mentioned clearly as a limitation and author's could discuss about the prognosis (citing other case reports), risks and if any recurrence is expected. This will be interesting for the readers. 

Comments on the Quality of English Language

English is fine. Minor editing and proof reading is necessary. 

Author Response

At first, we would like to thank you for your appreciated feedback helpful suggestions. Below you will find our responses to your comments.

Introduction

Comment  1) The statement about monostotic fibrous dysplasia being the most common form should be followed by specifying the site where it is most frequently reported. Additionally, it would be informative to mention the stage of life at which it most commonly presents.

Response 1) Thank you for pointing this out. We agree and therefore, we added sentence 30-31. We now mention "The latter presents most frequently in the femur and tibia, with lower frequency in the ribs [4]." Additionally, in sentence 32-33 we now describe "In general, patients present in the second or third decade of life, without gender predisposition [3]." 

Case report:

Comment 2) Regarding the figures, it would be beneficial to combine them into sets with figure panels. For example, all 3D constructed CT images could be consolidated into one figure with multiple panels. Similarly, the multiplanar CT images could be grouped. Figures 8 and 9, which show the resected tumor mass, can also be combined.

Response 2) Again, we fully agree. We now combined the multiplanar reformation of the preoperative computed tomography scan in Figure 2 (page 3). We combined the 3D reconstructions in Figure 3 (page 4). A panel was made for the resection specimens (Figure 5, page 6). 

Comment 3) Please include arrows to point to the spindle cell stroma and irregular trabecular bone in the histopathology slide, preferably in different colors.

Response 3) Thank you for mentioning. We have now added a black arrow, pointing towards two examples of the trabecular bone. Together with one of the co-authors, clinical pathologist, we discussed about the spindle cell stroma. The spindle cell stoma is, however, basically the entire content of the figure (except for the trabecular bone). 

We therefore hope that you agree that the following legend to the figure now adequate differentiates between the two cell types, especially where we mention "spindle cell stroma harboring irregular trabecular of woven boen". If not, then we will of course add a zoomed-in figure. 

"Figure 6. Histological specimen of the resected mass. Hematoxylin and eosin microscopy show the typical morphology of fibrous dysplasia of bone represented by spindle cell stroma harboring irregular trabeculae of woven bone (black arrows) with irradiating collagen fibers. The black bar in the lower left corner corresponds to 100µm." (page 7).

Comment 4) Including intra-operative pictures of the hemi-clamshell approach will be interesting for the readers.

Response 4) We now added figure 4 (page 5). This figure was added in order to give the readers insight in the semi-clamshell approach.

Comment 5) The case report ends abruptly without any follow-up data on the patient. How long was the patient followed up? Providing follow-up data for at least 6 months would be beneficial.

Response 5) Thank you again. We fully agree. We now added the follow-up up to 4 years (sentence 194-196). "Follow-up in this case is limited to 4 years postoperatively, which prohibits drawing conclusions on even longer follow-up, in particular on the evolution of the non-operated sites of the polyostotic FD."

Comment 6) It would be useful to include details about the post-operative rehabilitation protocols, including exercises, and how the patient tolerated them.

Response 6) We now describe the specific rehabilitation for this patient (sentence 142-145). As standard of care, the physiotherapist was consulted for the first four days. Rehabilitation was well tolerated. With first follow-up in the outpatient setting, patient was fully recovered. 

Comment 7) How long did it take for the patient to return to full function?

Response 7) We added sentence 144-145 (also mentioned above), where we mention that patient recovered within 3 months after surgery.

Discussion: 

Comment 8) If only a minimum follow up is available, it should be mentioned clearly as a limitation and author's could discuss about the prognosis (citing other case reports), risks and if any recurrence is expected. This will be interesting for the readers. 

Response 8) We fully agree that a longer follow-up was needed. We described it (at mentioned above). However, we still agree that "only" 4 years of follow-up is a limitation. Therefore, we added a limition section in our discussion (sentence 194-196).

Additonally, we added sentence 174-181 describing the recurrence rate: "Recurrence rate after surgical treatment of FD is rarely reported, with predominantly case reports with short follow-up presented in literature. For craniofacial FD, a systematic review showed recurrence rates of 4% up to 84%, with radical resections in non-syndromic cases and conservative surgery in syndromic cases, respectively [12]. In our case, the anatomical location facilitated radical resection and recurrence is therefore unlikely to occur. Thus, it might be hypothesized that recurrence is highly dependent on the anatomical location, surgical technique and whether FD occurs in a (non-)syndromic form."

Literature is scarce on the recurrence rate of FD as a "single modality". There are no large series. We therefore referring to a recent systematic review on craniofacial FD. The anatomical location of craniofacial FD, with often vital structure (mainly nerves) in close proximity, dictates surgical approach. Apparently, "debulking" is more frequent performed in those cases instead of radical resection. With that, recurrence rates are higher in those cases of debulking. Lastly, the review concludes that there is a difference in recurrence rate between syndromic and non-syndromic forms of FD. 

Altogether, we formulated the following hypothesis in the discussion (sentence 179-181).